Quantifying climate change impacts emphasises the importance of managing regional threats in the endangered Yellow-eyed penguin

Mattern Thomas t.mattern@eudyptes.net 1
Meyer Stefan 1
Ellenberg Ursula 2
Houston David M. 3
Darby John T. 4
Young Melanie 1
van Heezik Yolanda 1
Seddon Philip J. 1
1 Department of Zoology, University of Otago , Dunedin , New Zealand
2 Department of Ecology, Environment and Evolution, La Trobe University , Melbourne , Australia
3 Science and Policy Group, Department of Conservation , Auckland , New Zealand
4 Otago Museum , Dunedin , New Zealand
Gandini Patricia
Electronic publication date: 2017 May 16
Publication date: 2017
Volume: 5
Electronic Location ID: e3272
Received 2017 Feb 9; Accepted 2017 Mar 28
Copyright: ©2017 Mattern et al.
Copyright year: 2017
Copyright holder: Mattern et al.
License: This is an open access article distributed under the terms of the Creative Commons Attribution License, which permits unrestricted use, distribution, reproduction and adaptation in any medium and for any purpose provided that it is properly attributed. For attribution, the original author(s), title, publication source (PeerJ) and either DOI or URL of the article must be cited.
License URL: https://creativecommons.org/licenses/by/4.0/

Keywords: Climate change, Anthropogenic threats, Population modelling, Penguins, Species management, Demography, Survival rates, New Zealand, Conservation, Endangered species

Funding: University of Otago Research Grant This work was supported by an University of Otago Research Grant (issued to PJS). The funders had no role in study design, data collection and analysis, decision to publish, or preparation of the manuscript.

==============================
Climate change is a global issue with effects that are difficult to manage at a regional scale. Yet more often than not climate factors are just some of multiple stressors affecting species on a population level. Non-climatic factors—especially those of anthropogenic origins—may play equally important roles with regard to impacts on species and are often more feasible to address. Here we assess the influence of climate change on population trends of the endangered Yellow-eyed penguin (Megadyptes antipodes) over the last 30 years, using a Bayesian model. Sea surface temperature (SST) proved to be the dominating factor influencing survival of both adult birds and fledglings. Increasing SST since the mid-1990s was accompanied by a reduction in survival rates and population decline. The population model showed that 33% of the variation in population numbers could be explained by SST alone, significantly increasing pressure on the penguin population. Consequently, the population becomes less resilient to non-climate related impacts, such as fisheries interactions, habitat degradation and human disturbance. However, the extent of the contribution of these factors to declining population trends is extremely difficult to assess principally due to the absence of quantifiable data, creating a discussion bias towards climate variables, and effectively distracting from non-climate factors that can be managed on a regional scale to ensure the viability of the population.

Introduction

Climate change significantly alters the phenology and distribution of the world’s fauna and flora (Parmesan, 2006). Species with spatially limited distributions suffer particularly from climate-related change in their habitats, which can drive range shifts (e.g., Sekercioglu et al., 2008; Grémillet & Boulinier, 2009), range restrictions (Sexton et al., 2009) or, in the worst case, extinction (Thomas et al., 2004). Current climate predictions suggest that the pressure on ecosystems will continue to increase (IPCC, 2013), especially affecting species that occupy fragmented habitats. The spatial segregation of suitable habitat might preclude range shift adjustments and increase the risk of local extinctions (Opdam & Wascher, 2004).

For species conservation, this creates a daunting scenario. With resources for conservation often limited, the inevitability of climate change could be used as an argument against taking action to conserve species at locations that may become sub-optimal due to environmental change (Sitas, Baillie & Isaac, 2009). However, often cumulative anthropogenic impacts (e.g., habitat destruction, pollution, resource competition, accidental mortality) significantly add to—or even exceed—the impact of climate-related environmental change (Parmesan & Yohe, 2003; Trathan et al., 2015). While climate change is a global issue that is difficult to tackle at a regional scale, addressing local-scale anthropogenic factors can enhance species’ resilience to environmental change. Quantifiable data on climate variables are usually readily available through international and regional monitoring programmes (e.g., Kriticos et al., 2012), whereas this is generally not the case with other non-climate related data. Lack of monitoring or commercial interests often prevent the compilation of data (e.g., Chen, Chen & Stergiou, 2003; Mesnil et al., 2009) which may be relevant to species survival. This creates the risk of an analytical bias towards climate impacts, thereby distracting from and potentially understating non-climate threats.

The population status of New Zealand’s endemic Yellow-eyed penguin (YEP, Megadyptes antipodes) illustrates the complexity of this issue. YEP is a species of significant cultural and economic value for New Zealand (Seddon, Ellenberg & van Heezik, 2013). Particularly the tourism industry of the Otago Peninsula benefits from the presence of the birds with YEPs contributing more than NZ$100 mio annually to the local economy. (Tisdell, 2007). Ensuring the survival of the species is therefore not only a matter of ethical considerations, but also of economic importance.

With an estimated 1,700 breeding pairs the YEP is one of the rarest penguin species world-wide (Garcia Borboroglu & Boersma, 2013). Compared to other penguins, the YEP’s distributional range is fairly limited. About 60% of the species’ population is thought to inhabit the sub-Antarctic Auckland and Campbell Islands, while the remaining ∼40% breed along the south-eastern coastline of New Zealand’s South Island (Seddon, Ellenberg & van Heezik, 2013). Genetic analyses revealed that there is virtually no gene flow between the sub-Antarctic and mainland YEP populations (Boessenkool et al., 2009a).

While little is known about the sub-Antarctic populations, mainland YEPs have received considerable scientific attention. The first comprehensive studies of breeding biology and population dynamics were carried out in the first half of the 20th century by Lance Richdale (Richdale, 1949; Richdale, 1951; Richdale, 1957). Interest in the species waned after Richdale’s retirement from active research, but was rekindled in the late 1970s (Darby, 1985). Regular monitoring of some breeding sites commenced in the early 1980s, and was expanded and intensified following a catastrophic die-off affected breeding adult penguins on the Otago Peninsula in the austral summer of 1989–90 (Efford, Spencer & Darby, 1994). Parts of the population have been monitored without interruption since 1982 resulting in a data set spanning more than three decades (Ellenberg & Mattern, 2012). A recent review of available information revealed that a steady decline of the population might have been masked by more intensive monitoring since the early 2000s (Ellenberg & Mattern, 2012).

Most New Zealand penguin species including YEPs are believed to have undergone significant population declines in the past century, with climate change suspected to be playing a major role (e.g.,Cunningham & Moors, 1994; Peacock, Paulin & Darby, 2000). At the same time, penguin populations are exposed to numerous anthropogenic threats (Trathan et al., 2015). Climate variables and anthropogenic influences create a complex mix of factors that make it challenging to decipher the causation of population developments.

Using population data recorded between 1982 and 2015 from one of the YEP’s mainland strongholds, we developed a population model that integrates observed population changes with key climatic variables. While climate data are readily available as continuous data sets, data on anthropogenic factors are often sparse or of low temporal and spatial resolution which inhibits quantitative analysis. We assess to which extent population trends can be attributed to climate change so as to highlight and discuss the likely importance of other, not readily quantifiable but more manageable threats.

Methods

Species information

The IUCN Red list classifies Yellow-eyed penguins as “Endangered” (BirdLife International, 2016), and they are listed as “Nationally Vulnerable” under the New Zealand Threat Classification System (Robertson et al., 2013). The three main subpopulations are estimated to range between 520 and 570 breeding pairs on the Auckland Islands, 350–540 pairs on Campbell Island, and 580–780 pairs along New Zealand’s south-eastern coastlines and Stewart Island (Seddon, Ellenberg & van Heezik, 2013). On the mainland, the Otago Peninsula represents the species’ stronghold where numbers of breeding pairs in the past three decades have been as high as 385 in 1996, but have steadily declined over the last 20 years to only 108 pairs in 2011 (Ellenberg & Mattern, 2012).

Yellow-eyed penguins breed in the austral summer (September–February) so that their annual breeding period spans the turn of the calendar year. Socialising and courtship in July marks the onset of a new breeding season that ends in March/April with annual moult and subsequent replenishing of resources in preparation for the next breeding season (Seddon, Ellenberg & van Heezik, 2013). Hence, we used austral year (i.e., July to June) to calculate means and for summarising annual statistics of demographic and environmental parameters.

Study sites

The Otago Peninsula penguin population has received considerable scientific attention in the past century, with Richdale conducting his seminal population research between 1936 and 1954 (Richdale, 1949; Richdale, 1951; Richdale, 1957), followed by a string of projects from the 1980s onwards addressing many aspects of the Yellow-eyed penguin’s biology including phylogeny (e.g., Boessenkool et al., 2009b), breeding biology (e.g., Darby, Seddon & Davis, 1990), diet (e.g., van Heezik, 1990), foraging ecology (e.g., Mattern et al., 2007), and conservation (e.g., Ellenberg, Mattern & Seddon, 2013). While Richdale conducted most of his work at Kumo Kumo Whero Bay, most of the recent research was carried out at the Boulder Beach complex (Fig. 1) which, as a result, has the longest ongoing population monitoring program and the most reliable data set available (Ellenberg & Mattern, 2012).

Figure 1 Overview of the breeding range of Yellow-eyed penguins.

Overview of the breeding range of Yellow-eyed penguins, detail of the Otago Peninsula with an aerial view of the Boulder Beach Complex (henceforth Boulder Beach) with outlines indicating the locations of the four main monitoring plots. The inset map also indicates Kumo Kumo Whero Bay, the location of the historic population study conducted from the 1930s to 1950s.

Population monitoring & Yellow-eyed penguin database (YEPDB)

Flipper banding of Yellow-eyed penguins commenced at Boulder Beach in the 1970s and by the mid-1980s the majority of the local population was marked. Annual nest searches were conducted to determine number of breeders and repeated nest checks provided information on bird identity and reproductive success (Darby, 1985). After a catastrophic adult die-off during the 1989 breeding season (Gill & Darby, 1993), monitoring was intensified to include 60% of the known South Island breeding sites (Seddon, Ellenberg & van Heezik, 2013). The Yellow-eyed penguin database (YEPDB) was created in the early 1990s (Efford, Spencer & Darby, 1994) and is maintained by the New Zealand Department of Conservation (DOC) which also maintains the YEP monitoring and banding program. While the use of subcutaneous transponders has been introduced in the monitoring population, DOC still maintains flipper bands as primary marking method for a transitional period to ensure data consistency can be maintained before phasing out banding.

At the time of writing, the database contained banding records for 13,788 penguins (date range: 1973–2013), and 9,006 nest records (range: 1979–2014). It also holds information on incidental penguin recoveries or sightings outside the breeding season; however, these recovery data are patchy and were deemed too unreliable for analysis.

Data

Demographic data

Nearly one third of all banding records (n = 3,733) and nest records (n = 2,342) originate from Boulder Beach (Fig. 1) providing consistent, uninterrupted monitoring data for our analyses. While monitoring commenced in the late 1970s, first complete data sets are available from 1982 onwards, although for the first season there are only records of six nests.

Data were extracted from YEPDB as a series of SQL queries. Population numbers were retrieved from the table holding nest records. Number of breeding adults was calculated by multiplying the number of nests by two; number of fledglings is the sum of chicks fledged from all nests, and number of new breeders represents the sum of all adults that were recorded for the first time as breeders. Where possible we determined age of breeding birds per year by querying their banding details; age is unknown for birds banded as adults (ca. 15% of all banded birds).

To estimate demographic parameters, we first extracted ID numbers for individuals banded at the Boulder Beach complex since 1982. Secondly, we identified the years in which each bird was recorded as a breeding adult in the nest record table. Finally, we compiled the information from both database queries into a table where each column represented a nest year and rows comprised encounter histories for each individual. Birds had to miss at least two consecutive breeding seasons before being defined as dead or senescent. In a small number of cases a bird was not recorded as a breeder for three or more consecutive years before remerging as a nest occupant, but this applied to less than 1% of all birds.

Environmental data

We obtained monthly averages for selected climatic variables deemed likely to have an influence on demographic parameters (Table 1). The National Climate Database (CliFlo, http://cliflo.niwa.co.nz) has kept records from weather stations in Dunedin and the Otago Peninsula continuously since the early 20th century. Austral annual means were calculated for each parameter (i.e., July–June) as well as for the months March–May, which covers the penguins’ annual moult and post-moult periods. During this time birds are particularly susceptible to environmental perturbations due the increased energy requirements for feather replacement (Croxall, 1982). Data on local sea surface temperatures (SST) were obtained from the Portobello Marine Laboratory (University of Otago) which holds a near continuous time series of daily measurements dating back to January 1953. We calculated the monthly SST anomaly by subtracting monthly means from the average value calculated from all monthly means ranging from January 1953 to December 2014; annual SST anomaly is the mean of monthly SST anomalies for the corresponding year. To examine for potential lag effects of SST anomaly on prey availability (Beentjes & Renwick, 2001), we also examined SST anomalies shifted backwards in time by one and two years.

Table 1 Description of basic environmental parameters used for the development of a YEP population model.

Parameter	Shorthand	Station	
Total rainfall (mm)	total_rainfall	Southern Reservoir (National Climate database, CliFlo ID 5400)	
Wet days—Number of days With 1 mm or more of rain (days)	wet_days	Southern Reservoir (5400)	
Maximum 1-day rainfall—9 am to 9 am local time	max_1day_rain	Dunedin, Musselburgh (5402)	
Mean air temperature	mean_air_temp	Dunedin, Musselburgh (5402)	
Mean daily minimum air temperature	daily_min_temp	Dunedin, Musselburgh (5402)	
Days of wind gusts ≥ 33 Knots	days_wind_gusts_33	Dunedin, Musselburgh (5402)	
Sea surface temperature anomaly	sst_anomaly	Portobello Marine Lab, University of Otago	

Population model

We estimated adult survival and fledgling survival by developing a Bayesian mark-recapture (MR) model that incorporated effects of climate parameters. Chicks are only banded shortly before fledging, so that the MR model could not consider hatchlings that died before they were marked (i.e., chick survival). Hence, fledgling survival was adjusted by incorporating the proportion of chicks fledged to chicks hatched. We modelled survival in any year as a random process ranging around a mean of zero within the bounds of a total temporal variance. This allowed us to determine the relative importance of each climate covariate in terms of percentage of total variance explained (Grosbois et al., 2008). For models with covariates explaining at least 20% of the total variance, we estimated posterior model probabilities using Gibbs Variable Selection (GVS, Tavecchia et al., 2016)

Subsequently, we modelled YEP population dynamics via a female-only model assuming a birth-pulse population (Tang & Chen, 2002). The effect of environmental factors on the population growth rate was examined by using fixed survival rates (means) within the population model, allowing it to approximate the deterministic population growth rate between 1982 and 2014. Similarly, we estimated the population growth rate by changing mean survival rates corresponding to low SSTs that were measured from 1982 to 1996, and high SSTs characteristic for the time period from 1997 to 2015. Finally, we projected future populations by running a series of stochastic projections that used a range of survival rate estimates (i.e., omitting years with increasing uncertainty in estimate validity) and predicted trends in influential environmental factors.

Detailed descriptions of all modelling procedures are provided as Supplemental Information 1.

Comparison with historic population trends

Richdale (1957) provides comprehensive data on penguin demography allowing it to draw comparisons between historic and contemporary penguin numbers. We inferred population parameters from three tables. Table 67 (p147) provides direct information about the number of eggs laid and chicks fledged. Using number of eggs, we inferred the number of nests for the reported years by assuming only two-egg clutches were present. In Table 72 (p154), Richdale reports the percentage of surviving breeders of both sexes for each year, adjusted to the fractional format by dividing the reported values by 100. Finally, Table 62 (p138) provides clues about annual recruitment, which was calculated as proportion of new breeders each year. We omitted Richdale’s data for the 1936 season and for the seasons following 1949, as he noted less frequent monitoring and incomplete data sets for the initial and the latter years of his study (Richdale, 1957).

Results

Observed penguin numbers

Numbers of adult breeders at Boulder Beach fluctuated considerably between 1982 and 2015 (Fig. 2). Immigration of birds that had been banded outside Boulder Beach was a rare occurrence throughout the study period (mean proportion of immigrants per year 1982–2015: 2.7 ± 2.2%). If birds banded as breeders are considered to have come from other breeding sites, the median immigration is similar (2.0%) although three years (1991, 2010 and 2012) would stand out where unbanded adults made up 11, 10 and 8% of the breeding population, respectively. An apparent rise in penguin numbers at the beginning of the monitoring period (i.e., 1982–1985) reflects increasing monitoring effort. Reduced monitoring effort may explain the drop in numbers after 1985–86; two areas were not monitored in several years (A1: 1986–1989; Highcliff: 1989). Both areas account for 46 ± 4% of penguin counts (1990–2015), so that true penguin numbers in 1989 were likely considerably higher than the database would suggest. Breeder numbers in the two areas monitored in 1989 (Midsection; Double Bay) dropped by 62% in the following season (1989: 74 birds, 1990: 28 birds) when the population was affected by a catastrophic adult die-off. The population recovered between 1990 and 1996 to reach levels comparable to those observed in 1985. The 1996 season had the highest numbers of breeders recorded at Boulder Beach (n = 242) and represents a turning point for the population. Subsequently penguin numbers reached a low of 104 breeders in 2002, with losses compounded by another adult die-off event occurring in the 2001 season. Between 2002 and 2012 the population fluctuated between 100 and 150 breeders without any apparent trend before another drastic decline in numbers began in the years following a third adult die-off event at the end of the 2012 season. The steepest drop in numbers (41%) recorded since 1989 occurred between 2013 (128 breeders) and 2014 (76 breeders). In 2015, only 58 breeding penguins were recorded, which translates to a 76% decline in numbers since 1996.

Figure 2 Observed penguin numbers at Kumo Kumo Whero and Boulder Beach.

Observed penguin numbers at Kumo Kumo Whero 1937–1948 (from data published in Richdale 1957, see ‘Methods’ for details) and at the Boulder Beach complex 1982–2015 as extracted from the Yellow-eyed penguin database. ‘New breeders’ represents the portion of all ‘breeding adults’ that were recorded as breeders for the first time. Red arrows indicate years with observed die-off events affecting adult breeders. Note that as some sections of the Boulder Beach complex were not monitored in all years, data for the years 1986–1989 were adjusted by adding the mean proportion these areas contributed to the total count in all other years.

Number of chicks that fledged each year generally followed the trends observed for adults (Fig. 2). However, significant variation between 2003 and 2010 reflects a series of years with poor breeding success followed by better reproductive output in the following year. Numbers of new breeders showed a similar albeit weakened pattern delayed by 5 years: starting in 2004, numbers of new breeders seem to mimic those of fledglings beginning in 1999.

Age of breeding birds ranged between 8.4 years (1984) and 14.9 years (1990, mean: 12 ± 1.4 years, Fig. 3). Between 1990 and 2015 the average age of returning breeders showed a slightly decreasing trend from around 14 to 11 years (Pearson correlation ρ =  − 0.307, t24 =  − 1.5781, p = 0.13). At the same time, average age of new breeders dropped significantly from more than 10 years in the 1990s to only 4 years in 2015 (ρ =  − 0.796, t24 =  − 1.5781, p < 0.001). The average age of new breeders increased steeply after both the 1989 and 2001 adult die-offs (Fig. 3) indicating a substantial pool of older non-breeders ready to recruit following the disappearance of established breeders. No such spike is apparent after the 2012 die-off suggesting that the pool of older recruits has dried up over the last decade.

Figure 3 Age of breeding Yellow-eyed penguins.

Average age of breeding Yellow-eyed penguins active at Boulder Beach between 1982 and 2015. Red arrows indicate years with observed die-off events affecting adult breeders.

Demographic estimates from the mark-recapture model

The MR model without covariate revealed a fledgling survival rate of 0.12 (95% CRI [0.08–0.19]) in chicks (Table 2). The survival of adults was 0.87 (95% CRI: [0.83–0.90]). Throughout the study period (1982–2014), fledgling survival varied 2.56 times more than adult survival (95%, CRI [1.03–6.45]) (Table 2).

Table 2 Parameter estimates from the Bayesian mark-recapture model. Φ indicates estimated annual survival rates, σ2 stands for the temporal variance of the stage-specific annual survival.

Refer to ESM1 for details.

		Credible interval	
Parameters	Median	2.5%	97.5%	
Φ¯chicks	0.124	0.077	0.189	
σchicks2	1.877	1.001	3.847	
σchicks2 (on probability scale)	0.021	0.009	0.065	
Φ¯adults	0.872	0.832	0.904	
σadults2	0.732	0.414	1.398	
σadults2 (on probability scale)	0.009	0.005	0.021	

Years with increased wind activity had a positive effect on fledgling survival, whereas the effect of higher than normal SST was negative; both covariates explained 33.2% of the variance (Table 3A, ID 1 & 2). Similarly, SST anomaly during the first three months after fledging as well as in the previous year both had a negative effect on survival, explaining 24.8% and 17.4% of the variance (IDs 3 & 4), while increased wind activity in the months after fledging had a positive effect on fledgling survival (16.5% of variance explained, ID 5). Furthermore, years with above average air temperatures had a negative effect on fledgling survival, explaining 12.4% and 15.4% of the variance (IDs 6 & 7).

Table 3A Estimated effect size for fledgling survival (βfledlings).

Note, that negative values resulted from models that estimated slightly higher (or less precise) variance in fledgling survival, as it would result for the model without covariate. Except for covariate 2, 3, 4 and 15 all variables were standardized before fitted to the MR model.

			Credible interval		
ID	Covariate	Median	2.5%	97.5%	PVE	
1	days_wind_gusts_33_annual	0.850	0.377	1.329	33.2	
2	sst_anomaly_austral	−1.967	−3.148	−0.964	33.2	
3	sst_anomaly_minus_1yr	−1.516	−2.649	−0.392	24.8	
4	sst_anomaly_mar_june	−0.970	−1.845	−0.111	17.4	
5	days_wind_gusts_33_mar_may	0.696	0.198	1.241	16.5	
6	daily_min_temp_annual	−0.644	−1.190	−0.143	15.4	
7	mean_air_temp_annual	−0.590	−1.167	−0.102	12.4	
8	daily_min_temp_mar_may	−0.303	−0.829	0.204	0.7	
9	mean_air_temp_mar_may	−0.304	−0.850	0.190	0.2	
10	total_rainfall_may_may	−0.254	−0.823	0.296	−2.3	
11	max_1day_rain_mar_may	−0.250	−0.835	0.318	−3.3	
12	total_rainfall_annual	−0.260	−0.841	0.316	−3.5	
13	max_1day_rain_annual	−0.167	−0.738	0.394	−5.2	
14	wet_days_mar_may	−0.141	−0.702	0.431	−5.6	
15	sst_anomaly_minus_2yr	−0.217	−1.451	1.045	−5.6	
16	wet_days_annual	0.073	−0.461	0.623	−6.7	
Notes.

PVE percentage of variance in fledgling survival explained by each covariate

In adults, SST had the greatest effect on the survival rate, explaining 36.8% of the variance (Table 3B, ID 1). The relationship of adult survival and SST becomes apparent when the deviation of annual adult survival from the median survival rate is plotted against SST anomaly (Fig. 4). In periods with cooler than usual SST, adult survival was high (e.g., 1990–1996), whereas warm periods were characterized by lower adult survival. The same was true for air temperature. Warmer years were associated with reduced adult survival; air temperature-related covariates explained 34.4% of the variation in adult survival (Table 3B, IDs 2 & 3).

Table 3B Estimated effect size for adult survival (βadults).

Note, that negative values resulted from models that estimated slightly higher (or less precise) variance in adult survival, as it would result for the model without covariate. Except for covariate 1, 4, 5 and 14 all variables were standardized before fitted to the MR model.

			Credible interval		
ID	Covariate	Median	2.5%	97.5%	PVE	
1	sst_anomaly_austral	−1.267	−1.925	−0.631	36.8	
2	mean_air_temp_annual	−0.529	−0.817	−0.251	34.4	
3	daily_min_temp_annual	−0.516	−0.796	−0.227	34.4	
4	sst_anomaly_mar_june	−0.808	−1.329	−0.310	26.2	
5	sst_anomaly_minus_1yr	−1.056	−1.719	−0.406	25.7	
6	days_wind_gusts_33_annual	0.377	0.075	0.690	16.5	
7	days_wind_gusts_33_mar_may	0.350	0.052	0.666	12.8	
8	daily_min_temp_mar_may	−0.214	−0.537	0.088	2.0	
9	total_rainfall_may_may	−0.146	−0.461	0.193	1.0	
10	mean_air_temp_mar_may	−0.181	−0.513	0.140	−0.3	
11	wet_days_mar_may	−0.113	−0.434	0.207	−1.0	
12	max_1day_rain_mar_may	−0.098	−0.416	0.234	−2.0	
13	max_1day_rain_annual	0.112	−0.206	0.435	−2.2	
14	sst_anomaly_minus_2yr	−0.055	−0.867	0.720	−3.1	
15	wet_days_annual	0.064	−0.275	0.393	−3.8	
16	total_rainfall_annual	0.057	−0.268	0.391	−4.0	
Notes.

PVE percentage of variance in adult survival explained by each covariate

Figure 4 Adult survival and SST anomalies.

Top graph: local Sea Surface Temperature anomalies recorded at Portobello Marine Lab, Otago Peninsula, between 1953 and 2016. Bottom graph: detail of SST anomalies 1980–2016 and associated deviance (black line: mean; grey area: 95% credible interval) in survival of adult Yellow-eyed penguins as determined from a MR recapture model.

Refitting the MR model with the two most influential explanatory covariates each for fledging and adult survival, and subsequent assessment of posterior model probability, ranked highest the model where both chick and adult survival were fitted to the single covariate SST anomaly (Table 4).

Table 4 Results of the gibbs variable selection.

0 and 1 indicate whether each covariate is not included or included in the model, respectively. The MR considers covariates ‘sst_anomaly_austral’ for fledgling (A) and adult survival (C), ‘days_wind_gusts_33_annual’ (B) and ‘mean_air_temp_annual’ (D). For a detailed description of the GVS refer to ESM4.

	Model configuration		
	Fledgling survival	Adult survival		
Mi	A	B	C	D	p(Miy ¯)	
1	1	0	1	0	0.42	
2	0	1	1	0	0.13	
3	1	1	1	0	0.12	
4	1	0	0	1	0.09	
5	0	1	0	1	0.06	
6	0	0	0	0	0.04	
7	1	0	0	0	0.03	
8	0	1	0	0	0.03	
9	1	1	0	0	0.02	
10	0	0	1	0	0.01	
11	0	0	0	1	0.01	
12	1	1	0	1	0.01	
13	0	0	1	1	0	
14	1	0	1	1	0	
15	0	1	1	1	0	
16	1	1	1	1	0	

Predictions for the adult female population

Using year-specific survival rates from the MR model generates predictions of numbers of adults that were similar to those determined during monitoring. For most years, the observation-based number of adult female YEPs and the 95% credible intervals for the predicted number of adult female YEPs overlapped (Fig. 5).

Figure 5 Population projections for Yellow-eyed penguins at Boulder Beach, Otago Peninsula.

Population projections for Yellow-eyed penguins at Boulder Beach, Otago Peninsula. The graphs show the observed (red line) and estimated (black line) number of female penguins, and associated 95% credible interval (grey area), as derived from the population model. The dashed vertical line indicates the last year used to parameterise the MR model and the starting year of the simulation. Population projections were modelled using survival rate estimates until 2012; beyond this year estimates get increasingly unreliable because these are based on data about individual absence from breeding rather than from reported mortalities (see ‘Methods’).

Based on a deterministic model (i.e., without temporal variance in survival rates) the population growth rate was 1.02 (95% CRI [0.98–1.06]) per year throughout the entire study period. For the time period when SST was below average (1982 to 1996, Fig. 4) the population showed an increasing trend with a growth rate of 1.038 (95% CRI [0.99–1.080], Fig. 6). However, from 1996 onwards an ongoing period of mainly warmer than normal SST went along with a growth rate of 0.94 (95% CRI [0.90–0.98]) indicating a population decline (Fig. 6).

Figure 6 Probability density functions for growth rates.

Probability density functions for deterministic annual population growth rates derived from survival rates that were rescaled for periods of cooler (1982–1996) and warmer (1996–2014) than average sea surface temperatures.

Future projections

Based on projections of increasing SST at a rate of 0.02 °C per year in the next decades (Oliver et al., 2014), the penguin population at Boulder Beach will continue to decline. Stochastic simulations using the most reliable estimates for adult survival (1982–2012) suggest that the number of adult female penguins will drop below 10 individuals by 2,048 (Fig. 5). If the recent poor breeding years 2013–2015 are included this negative trend gets progressively worse. Including adult survival rates estimated for 2015, the mean projection predicts YEPs to be locally extinct by 2043.

Discussion

Numbers of Yellow-eyed penguins at Boulder Beach have declined since 1996 (Figs. 5 and 6). The local population seemed to experience a reprieve from this decline in the first decade of the new millennium, despite unfavourable climatic conditions at that time. This might have been driven by a temporary reduction in other, non-climate negative impacts, the nature of which remain unclear due to a lack of data.

The ages of breeding penguins provide some explanation about the underlying mechanics of the population decline. In the years following the 1989 and 2001 adult die-offs, the average age of new breeders recruited into the population was substantially higher than in the years prior to the events. All of these birds were locally banded individuals, which suggests that there was a pool of older, previously unpaired birds which replaced experienced breeders that had died during the event. After the 2012 die-off, the mean age of new breeders reached an historic low (4.1 years, Fig. 3). Hence, old breeders that had lost their partner now paired up with younger penguins indicating that the pool of older non-breeders available to replace lost birds had disappeared. This is supported by the number of recruits reflecting the marked variation in fledgling numbers with a 5-year-lag (Fig. 2). It appears that since the turn of the century, penguins recruit into the breeding population at the earliest possible opportunity. This likely has negative effects on breeding performance since in seabirds age is an important determinant for foraging success (e.g., Daunt et al., 2007; Zimmer et al., 2011) and subsequently reproductive success (e.g., Limmer & Becker, 2009; Nisbet & Dann, 2009). The decline in the mean age of new breeders in recent years indicates that more inexperienced birds are recruiting as breeders, and possibly explains the overall deteriorating reproductive success.

When the 2012 die-off of adult breeding birds occurred, penguin numbers were less than 60% of what they had been in the mid-1990s (Fig. 2). While the penguin population showed a remarkable recovery after the 1989 event this did not happen following 2012; instead numbers have continued to decline. The most apparent differences following the two die-offs are the trends in ocean temperatures with a cooler-than-normal period in the first half of the 1990s whereas SST has been almost continuously higher than the 1953-2014 average since the late 1990s (Fig. 4).

Sea surface temperature effects

Sea surface temperature explained 33% of the variation in observed population trends. Hence, SST has an important influence on YEP population trends. Years with warmer than usual SST result in reduced adult survival, whereas the reverse is true when SST is cooler.

Variation in SST likely influences the abundance and quality of YEP prey. In Little penguins (Eudyptula minor) breeding on the Otago Peninsula, climatic fluctuations—and connected to this, ocean temperatures—were found to affect prey composition (Perriman et al., 2000). Little penguins are generalist foragers that take a variety of pelagic prey (Dann, 2013), most likely a beneficial trait in relation to climate related change in resource abundance (Thuiller, Lavorel & Araújo, 2005). YEPs on the other hand, are principally benthic foragers (Mattern et al., 2007) that feed predominantly on demersal species (e.g., van Heezik, 1990; Moore & Wakelin, 1997; Browne et al., 2011). Although this specialisation reduces competition for pelagic prey with the abundant marine avifauna in New Zealand (Mattern et al., 2007), it comes at the cost of reduced behavioural flexibility to respond to changes in prey distribution or abundance (e.g., Browne et al., 2011; Mattern et al., 2013).

Temperature affects the annual biomass of many fish species in New Zealand (Beentjes & Renwick, 2001). Warmer than normal conditions negatively affect spawning in fish, reducing subsequent recruitment (e.g., Takasuka, Oozeki & Kubota, 2008). Abundance of the demersal Red cod (Pseudophycis bacchus), historically an important prey species for YEP from Boulder Beach (van Heezik, 1990; Moore & Wakelin, 1997), shows a strong correlation to SST fluctuations, albeit with a lag of 14 months (Beentjes & Renwick, 2001). At Boulder Beach, a reduction in body mass of breeding YEPs in 1985 when compared to 1984 was associated with lower quantities of red cod taken (van Heezik & Davis, 1990). 1983 featured cooler than normal SST (mean monthly SST anomaly: −0.73), while 1984 temperatures were above average (SST anomaly: 0.17). As such the lagged correlation between SST and red cod abundance reported by Beentjes & Renwick (2001) also seems to be manifested in penguin body condition. This explains the relative importance of the corresponding covariate (i.e., sst_anomaly_minus1year) for survival rates (Tables 3A and  3B) and corresponds to findings of a previous analysis of climate variables on YEP numbers (Peacock, Paulin & Darby, 2000).

However, model selection showed an even stronger direct SST effect (Table 4). Ocean temperatures play an important role in the spatial distribution of fish populations (Beentjes et al., 2002). Warmer than usual SST is often an indication of increased stratification of the water column where a layer of warmer water sits on top of cooler water. This disrupts the benthic-pelagic coupling, i.e., mixing processes that regulate nutrient flow between benthos and surface waters (Jones et al., 2014). Land run-off has been identified as a major source of nutrients for the South Otago continental shelf, which results in higher near-surface nutrient concentrations (Hawke, 1989), so that vertical mixing is likely of crucial importance for benthic productivity and subsequent prey abundance in the penguins’ home ranges. Penguin foraging conditions are likely compromised under stratified, warm-water conditions.

The three major die-offs of adult penguins (seasons 1989–90, 2001–02, and 2012–13) all occurred in years with higher than normal SST suggesting that stratification might have more severe impacts than can be explained by the disruption of nutrient fluxes alone.

SST and relevance of die-off events

Die-off events do not seem to be related to prey availability; body condition of adult penguins examined during the 1989 event did not indicate malnutrition (Gill & Darby, 1993). The cause of mortality could not be identified although necropsies after the 2012 die-off indicated it to be toxin related (Gartrell et al., 2016). Harmful algal blooms (HAB) that are known to have negative impacts on other penguin species (Shumway, Allen & Boersma, 2003) were suspected to be involved in the die-offs as well (Gill & Darby, 1993). Yet water samples taken along a transect through the penguin’s known foraging ranges found no evidence for the presence of harmful algae (P Seddon, 2013, unpublished data). Tests for the presence of marine biotoxins in freshly dead birds were negative (Gartrell et al., 2016). Moreover, it seems unlikely that a HAB would selectively affect only one seabird species (Shumway, Allen & Boersma, 2003); no other unexplained seabird deaths occurred during any of the die-offs. Only bottom foraging YEPs were affected suggesting that the distribution of a toxin was probably limited to the near-seafloor region. Stratification and the disruption of vertical mixing potentially would contribute to a concentration of toxic components at the sea floor. While the origin or exact nature of the toxin remains unclear, it could be related to technical malfunctions that occurred at the time at Dunedin’s sewage treatment plant, which discharges at the seafloor about 1.5 km from the shore and ca. 5 km upstream from Boulder Beach (Dunedin City Council, 2013, unpublished data).

Although the cause of die-off events remains a matter of speculation, their relevance for population trends is closely tied to prevalent environmental conditions following these events. The 1989 die-off, which removed about 50% of penguins from the breeding population (Efford, Spencer & Darby, 1996) was followed by a six year period of population recovery, likely aided by cooler than normal SST (Fig. 4). The next die-off event occurred at Boulder Beach in 2001 (A Setiawan, pers. comm., 2004) and reduced the local population by nearly 40%. Following this event, the population showed no sign of recovery during a prolonged period of warmer-than-normal SST that began in 1998 and prevails until today. The associated reduced adult survival explains the lack of recovery in the penguin population. Consequently, the 2012 die-off had a cumulative effect, further reducing the population to its lowest level on record.

With projected SST increases over the next decades it seems doubtful that optimal marine conditions supporting the recovery of YEPs will occur in the future. Hence, future die-off events will be increasingly critical for penguin numbers. However, sea surface temperatures only explained about one third of the variation in survival rates. This means that other factors also play important roles for YEP population dynamics.

Other climate factors

Daily minimum air temperature is a proxy for prevailing temperature regimes, where a higher average minimum temperature indicates warmer years. Air temperature could simply be a covariate of SST and affect penguin survival through the mechanisms suggested above. In addition, air temperatures recorded during the moult (March–May) negatively affected adult survival probably as a result of hyperthermia. Little penguins in Australia suffer increased adult mortality when exposed to higher temperatures when moulting (Ganendran et al., 2015). However, there is no evidence for comparable temperature-related mortality events in YEP

Frequency of days with strong winds had a positive influence on fledgling survival. Wind aids oceanic mixing processes and thereby can become a driver for foraging success in penguins (Dehnhard et al., 2013). Wind generally acts as an antagonist to SST-related stratification effects, creating enhanced foraging conditions for penguins thereby increasing the survival chances of inexperienced fledglings.

Non-climate factors

In this study we were able to use comprehensive data to test the influence of a wide range of climate related factors on the population developments of Yellow-eyed penguins from Boulder Beach. Yet only about a third of the variation in penguin numbers can be explained by climate factors. Hence, it is clear that other, non-climate factors significantly affect penguin survival rates. While several of these factors are well known, it is impossible to examine their impact on the penguin population in a modelling context due to a lack of any quantifiable data. At the same time, unlike the effects of climate change, at least some of these non-climatic factors could be managed on a regional scale to enhance the species’ chance for survival. Therefore it is imperative to discuss some of these non-climate factors to avoid an undue focus on only the quantifiable factors (i.e., those driven by climate change) and direct conservation management towards measures that can ensure persistence of the Yellow-eyed penguin on the New Zealand mainland.

Fisheries interactions

Potential impacts of incidental bycatch in gill net fisheries (Darby & Dawson, 2000) and alteration of the penguins’ benthic foraging habitat by bottom fishing activities (Ellenberg & Mattern, 2012; Mattern et al., 2013) could not be quantified because data on gill net fisheries supplied by the Ministry of Primary Industries (NZ Ministry Of Primary Industries, Official Information Act Request OIA12-397) proved to be spatially coarse and temporally limited, with approximate locations of gill net fishing events specified only from 2006 onwards. Provided data on bottom fishing effort only covered the years 2000–2012 and originated from vessels operating outside the penguins’ ranges (OIA12-460).

The impact of single fisheries interactions might have a much greater effect on penguin numbers than annual fishing statistics would suggest. There are reports of multiple YEP killed in a single gill net haul (Ellenberg & Mattern, 2012) and reported bycatch incidents in gill net fisheries have been as high as 12 cases per year, many of which affected YEPs from the Otago Peninsula (Darby & Dawson, 2000). Currently, less than 2% of gill net effort in New Zealand is being independently observed (Richard & Abraham, 2015); this lack of observer coverage prevents reliable quantification of bycatch mortality. Yet it stands to reason that incidental fisheries mortality is an important factor affecting penguin survival rates and, hence, population trends.

Impacts of bottom fishing activities on YEP survival are even more difficult to quantify. Bottom trawling and dredge fisheries can substantially alter the benthic environment, reducing biodiversity, and prey abundance and quality for YEPs (Ellenberg & Mattern, 2012). Low quality prey were brought ashore by YEPs on Stewart Island, which had home ranges that apparently avoided the vast areas of potential habitat subject to intensive oyster dredging (Browne et al., 2011; Ellenberg & Mattern, 2012). On the Otago Peninsula, some penguins forage along straight-line paths following bottom trawl scrape marks, searching for scavenging prey that appears to be inadequate food for young chicks (Mattern et al., 2013).

Disease outbreaks

In the past decade several breeding seasons saw the occurrence of diphtheritic stomatitis, a secondary infection negatively affecting chick survival (Houston & Hocken, 2005). We could not test the effects of such disease outbreaks on population trends, because the YEP database does not facilitate quantitative storage of disease-related data. Diphtheritic stomatitis only affects chicks which generally survive when older than two weeks (Alley et al., 2016). Therefore, the disease is unlikely to have a lasting effect on population trends as it does not affect adults which are critical for the maintenance of a stable population (Benton & Grant, 1999). Although YEPs are subject to exposure to avian malaria parasites (Graczyk et al., 1995), observed infections are low, hence, avian malaria currently does not present a significant problem for the species (Sturrock & Tompkins, 2007). Avian pox which is caused significant mortality events in Magellanic (Spheniscus magellanicus) and possibly Gentoo penguins (Pygoscelis papua) has not been observed in YEP, although diphtheritic stomatitis may be the result of a secondary bacterial infection caused by a poxvirus (Alley et al., 2016).

Predators

Introduced terrestrial predators are one of the biggest challenges for native wildlife in New Zealand (Wilson, 2004). Mustelids (Mustela sp.), dogs (Canis lupus familiaris), and to a lesser extent cats (Felis catus) and rats (Rattus sp.) can impact on YEP (e.g., Alterio, Moller & Ratz, 1998; Ratz & Murphy, 1999), but it is very difficult to quantify these effects because direct evidence of predation is sparse. A five year study investigating the impact of feral cats on penguins on Stewart Island did not find any indication for predation events and concluded that starvation and disease were the main factor of mortality (King, 2008). On the mainland, predation by dogs or stoats appear to be very localised occurrences (Hocken, 2005). However, climate change may render this an increasing problem in the future (Tompkins, Byrom & Pech, 2013).

Predation by the native NZ sea lion (Phocarctos hookeri) has to date been limited to two female sea lions that were active between 1997 and 2005 (Lalas et al., 2007) that have since died (J Fyfe, pers. comm., 2014). More recently, a number of YEPs have been reported with injuries that were speculated to have been inflicted by Barracouta (Thyrsites atun). Considering that barracouta are smaller than adult YEPs (mean body lengths—barracouta: 55 cm, Fishbase.org 2016; YEPs: 65 cm, Seddon, Ellenberg & van Heezik, 2013) such injuries are at best an accidental consequence of penguins and fish targeting the same prey patch. Some external injuries might be the result of interactions with humans; in Australia, Little penguins (Eudyptula minor) have been injured and killed by water craft such as jet skis (Cannell et al., 2016), a recreational activity that has also been observed in the penguin landing zone at Boulder Beach (T Mattern, pers. obs., 2012).

Human impacts

The significance of human impacts in the form of deforestation of breeding habitat, capture by collectors, egging, and shooting of adults on the YEP population was highlighted early by Richdale (1951). While these impacts are no longer an issue, unregulated tourism has become an important threat at some Yellow-eyed penguin colonies and is reflected in reduced breeding performance and a steady decline of local penguin numbers (e.g., McClung et al., 2004; Ellenberg et al., 2007; Ellenberg, Mattern & Seddon, 2009).

‘Maladapted colonizer’ an oversimplification

Comprehensive analysis of ancient penguin DNA in recent years have revealed that the YEP is relatively recent colonizer originating from sub-Antarctic. The species is believed to have replaced a sister taxon Megadyptes waitaha after it was hunted to extinction by humans as recently as 500 years ago (Boessenkool et al., 2009b; Rawlence et al., 2015). In this light, the question was raised whether the species’ vulnerability to increasing ocean temperatures may in fact reflect a maladaptation for a warmer climate (Waters & Grosser, 2016). While evidence for a physiological relationship between ocean warming and survival rates in YEP is lacking, the specialized benthic foraging strategy renders the species particularly sensitive to environmental change (Mattern et al., 2007; Gallagher et al., 2015). With the various non-climatic factors discussed above all contributing to significant shifts across the entire benthic ecosystem, reducing the penguins’ struggle to a species-specific maladaptation for a warming climate clearly oversimplifies the matter.

Conservation implications

Stochastic simulations of future population trends for Yellow-eyed penguins at Boulder Beach, show that the population will continue to decline if current threats continue unabated. Global ocean temperatures are rising (IPCC, 2013); projections for the Tasman region until 2060 predict an increase in SST of up to 2 °C (Oliver et al., 2014), hence future climatic conditions will not be favorable for a recovery of the YEP population.

On the bright side, climate change-related pressure on YEP can likely be offset through control of the other more manageable factors negatively affecting population trends. This has already been demonstrated: positive YEP population growth during the 1940s, at a time when SST was strongly increasing in the Pacific to levels comparable to those recorded in the second half of the 1990s (Guan & Nigam, 2008), was attributed to a reduction in human impacts such as conversion of breeding habitat to farm land, establishment of road networks, road traffic and random acts of violence (Richdale, 1957). During World War II, when resources were directed towards the war effort, ‘man’s destructive agencies were practically negligible’ (Richdale, 1957, p157).

While climate change is a global phenomenon that is both inevitable and quantifiable, it is important to bear in mind its impact on species population trends is relative to other more regional factors, such as, in the case of penguins, fisheries, pollution, habitat destruction, introduced terrestrial predators, and human disturbance (Trathan et al., 2015). Managing local and regional factors can increase the resilience of species towards increasing pressure from climate change.

The virtual absence of quantifiable data to examine the effects of non-climate factors makes it difficult to provide evidence-based management recommendations and puts a potentially overbearing emphasis on climate change. However, these principally anthropogenic factors likely also explain significant portions of the variation in survival rates, so that the focus should be on improving our understanding and management of these impacts to enhance this species’ resilience to climate change.

Supplemental Information

Supplemental Information 1 ESM1—description of population model

Click here for additional data file.

Supplemental Information 2 ESM2—M-arrays

Click here for additional data file.

Supplemental Information 3 ESM3—Posterior distributions for all three prio configurations

Click here for additional data file.

Supplemental Information 4 ESM4—Correlation matrices for model parametrization

Click here for additional data file.

Supplemental Information 5 ESM5—gibbs variable selection

Click here for additional data file.

Supplemental Information 6 Mark-recapture & co-variate data

Click here for additional data file.

We would like to extend our gratitude to the many field workers and students involved in penguin monitoring that contributed data to the YEPDB. Special thanks are due to Bruce McKinlay, Jim Fyfe, and Jim Watts from the Department of Conservation Coastal Otago office for their efforts to maintain a regular monitoring schedule in the face of limited resources within the department.

Additional Information and Declarations

Competing Interests

Author Contributions

Data Availability

Yolanda van Heezik is an Academic Editor for PeerJ.

Thomas Mattern and Stefan Meyer conceived and designed the experiments, performed the experiments, analyzed the data, contributed reagents/materials/analysis tools, wrote the paper, prepared figures and/or tables, reviewed drafts of the paper.

Ursula Ellenberg, John T. Darby, Yolanda van Heezik, Philip J. Seddon, David M. Houston and Melanie Young contributed reagents/materials/analysis tools, reviewed drafts of the paper.

The following information was supplied regarding data availability:

The raw data has been supplied as a Supplementary File.

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
