# Peer review of "Quantifying climate change impacts emphasises the importance of managing regional threats in the endangered Yellow-eyed penguin"

_PeerJ, doi:10.7717/peerj.3272_

## Round 0.1 · original submission · Minor Revisions

This is a very interesting paper that uses an incredible database on Yelow-eyed penguins. Both reviewers agree it is generally very well written paper. Both of them make minor suggestions that will improve findings´interpretations. Take into account specially Rev. 1 comments on explaining how predation varies in relation to climate variation. Is there an increasing on avian diseases in warmer and wetter years? . Also occasionally the wording could be clarified in some sentences (see Rev. 2 suggestions).

·

Basic reporting

.

Experimental design

.

Validity of the findings

.

Additional comments

This is an excellent paper that Mattern et al. have pulled together using a long term data set on YEP to examining climate change. It is well written and argued. It is certainly worthy of publication. However, I think it would be improved with more attention to factors other than climate. My major question is can more be said that the impacts are not predator or disease related? Can the authors show that the control of predators has not made a significant change in adults or chick survival or in recruitment? As the paper states the data base does not allow him to say much about deaths but isn’t there something more that could be said about predation? Predation and land use is an important factors that needs more attention. How has what is in the data base about predation varied over the entire time period. Is there nothing on predation? Some predator control has been done in New Zealand since at least the 1980’s. . Does predator control show up as increased egg or chick survival? Is there no trend? This is key to know as lots of money is spent on predator control. Do stouts kill adults? At least I’d like to see that predation does not increase or decrease in relation to climate variation. Is avian pox detected and is it higher in warmer and wetter years? Nothing is mentioned about pox and that may be because it does not occur or has no been detected.

It’s an interesting story but I’d like a bit more to show the climate change story can not be explained by variation in predation or disease. Both predation and disease could be coupled with climate variation so I’d like more evidence that these variables are not key to the story.

This is a well written paper. It has been shown that penguins are impacted by increases in surface water temperature as Galapagos penguin quit breeding (Boersma 1978. Science 200: 1481:1483) so the authors may want to show this is a known relationship.

Reviewer 2 ·

Basic reporting

The paper is generally very well written.

Experimental design

The design and statistical treatment are generally sound.

Validity of the findings

I feel there is room for improvement with regard to data interpretation, with a more straightforward approach emphasising the most clear and important findings of the study - see below.

Additional comments

This study presents a detailed analysis of associations between variation in yellow-eyed penguin survival/recruitment, and environmental variation. The paper is generally very well written, and the statistical treatment is sound. My comments below are intended to improve the paper, and to help emphasize some of the most important findings which I feel the paper ‘undersells’ at present.

Occasionally the wording could be clarified – e.g. line 16:
“more often than not comprising only some of multiple stressors” – wording is convoluted - please simplify.

There is also a tendency to include statements that seem unclear and/or difficult to substantiate, and would be better restated more simply: e.g. lines 17-19:
“Non-climatic factors – especially those of anthropogenic origins – play equally if not more important roles with regard to impacts on species” – presumably the relative importance/impacts of climatic/non-climatic and human/non-human factors will vary hugely between taxa and regions.

Line 19 “relative influence” – relative to what?

Following the finding that 33-38% of variation in recruitment/survival can be explained by SST anomalies, there seems to be a subsequent suggestion that this result might be an artefact (e.g. lines 28-31 and elsewhere). Does the lack of good quantification of non-climatic ‘threats’ really create an analytical ‘bias’ (line 30) towards climate, artificially over-emphasising the importance of this variable? This statement seems to be unsubstantiated, and is not convincing. One could equally argue that including more detailed climatic data might well lead to this factor accounting for even more of the variation.

Similarly, L526-527: To describe the emphasis on climate as “potentially overbearing” seems strange given that SST has by far the greatest explanatory power in the analyses presented here. The authors seem to be going out of their way to play down the importance of their findings with regard to climate.

Indeed, it could be suggested that the importance of climate change affecting this species has tended to be under-emphasized until now. In summary, I feel it is important that the authors openly acknowledge the very real threats posed here by climate warming.

At the moment the study seems sometimes to be couched in a ‘vacuum’ of the very recent past, lacking broader context. It should surely be clarified early in the paper, and again in the discussion, that this cold-adapted species originated from the subantarctic, arrived in NZ during a cool-climate period at around 1500 AD (at the start of the Southern Hemisphere’s Little Ice Age, Waters et al 2017 J Biogeography; Rawlence et al 2015 Quaternary Science Reviews). This context might help to explain the inability of ‘large’ subantarctic M. antipodes to succeed further north where relatively ‘small’ mainland M. waitaha once lived. It also might explain why mainland NZ M. antipodes apparently remains particularly vulnerable to warming.

Line 341-366: There is extensive discussion that climate change might be affecting prey availability, and extensive consideration of numerous additional potential factors such as stratification, toxins, fishery impacts, predation, disease, tourism (lines 382-399, 442-503 etc), many of which appear to be ‘non-starters’, but very little mention of the far more straightforward scenario that warm temperatures on land and/or at sea might directly generate physiological stress, leading directly to mortality. Although there is a very brief acknowledgement of this possibility (hyperthermia; l420-422), I feel the potential direct, overarching importance of temperature is underexplored and underemphasised here.

---

## Round 0.2 · accepted · Accept

After reading your response to the reviewers and having reviewed the changes that were made to the paper I conclude that your Manuscript can be Accepted for publication